# Rooting Media and Biostimulator Goteo Treatment Effect the Adventitious Root Formation of *Pennisetum* 'Vertigo' Cuttings and the Quality of the Final Product

**Anna Kapczyńska** [1],*, **Iwona Kowalska** [2] , **Barbara Prokopiuk** [1] **and Bożena Pawłowska** [1]

1 Department of Ornamental Plants and Garden Art, University of Agriculture in Krakow, 29-Listopada 54, 31-425 Kraków, Poland; barbara.prokopiuk@urk.edu.pl (B.P.); bozena.pawlowska@urk.edu.pl (B.P.)
2 Department of Plant Biology and Biotechnology, University of Agriculture in Krakow, 29 Listopada 54, 31-425 Kraków, Poland; iwona.kowalska@urk.edu.pl
* Correspondence: anna.kapczynska@urk.edu.pl; Tel.: +48-12-662-52-49

**Abstract:** This study evaluated rooting of *Pennisetum* 'Vertigo' cuttings and development and the nutritional status of rooted plants. Cuttings of *Pennisetum* 'Vertigo' rooted in a perlite or peat medium were treated with Goëmar Goteo biostimulator as follows: (1) soaking (2) watering (3) spraying (4) no Goteo applied (control). Then, 83.3–100% of cuttings formed adventitious roots. The, 100% rooting was obtained for plants in perlite when Goteo spraying or watering was used for plants rooted in peat only after Goteo watering application. Cuttings rooted in perlite had 30% more roots and they were longer than in peat. Goteo watering of cuttings affected root elongation in both peat and perlite. Neither rooting media nor biostimulator treatment affected root dry weight (DW). Rooting medium after 2 months of pot cultivation had no effect on biometric features of plants, but those grown from cuttings rooted in peat had a higher fresh weight (FW) compared to those rooted in perlite. Plants developed from Goteo-treated cuttings were higher compared to the control plants. Goteo watering during rooting stimulated the formation of new shoots in the greenhouse cultivation. Plants from cuttings rooted in perlite had more Fe and Cu in their leaves, especially when they were Goteo-watered. Goteo increased P content in plants derived from biostimulator-watered cuttings, and K in plants from cuttings soaked in Goteo and rooted in perlite.

**Keywords:** rhizogenesis; marine algae; perlite; peat

## 1. Introduction

The concept of using ornamental grasses to create green areas has become an attribute of modern landscape architects, whose projects are implemented in home gardens as well as in multi-hectare urban areas [1,2]. Ornamental grasses have created a dynamic sector of floriculture where a wide range of new varieties is introduced each year. Market competition forces producers to act from the very beginning according to procedures that guarantee obtaining the best quality product. The genus *Pennisetum* Rich. (Poaceae) represents more than 80 grass species native to tropical and warm/temperate regions of the world [1]. Studies show that many of *Pennisetum* ornamental grasses are becoming popular—they are subjected to genetic testing to determine karyotype diversity and nuclear genome [3]. Breeding programs in recent years have resulted in obtaining the new *Pennisetum* hybrids for container and landscape applications [4]; one of them is *Pennisetum* 'Vertigo', product information: Graceful Grasses® Vertigo® *Pennisetum purpureum* 'Tift 8'. It is a trispecific ornamental hybrid (*P. purpureum*, *P. squamulatum*, *P. glaucum*) and it does not produce seeds; thus, its propagation must be carried out using vegetative methods. This plant can be treated as a perennial or annual grass depending on the hardiness

zone [5]. The reddish-purple color of mature leaves and rapid and intensive growth resulting in a large size (1.5 m) fully predisposes this cultivar for special cultivation as an ornamental plant both in containers and open space. Our observations have indicated a high demand for this type of plants which can be used in landscapes as an accent or background for other plants. Moreover, 'Vertigo' is a resistant variety to *Helminthosporium* leaf spot disease (adverse fungal symptoms observed on plants from the genus *Pennisetum*) [6]; thus, it can be a good alternative to sensitive ornamental species traditionally grown in landscaping. Competition of horticultural products is associated with the necessity of developing reproduction principles based on a comprehensive agricultural technology, including in vitro techniques [7]. Propagation of new cultivars is a complicated and long-term process. Thus, Goëmar Goteo biostimulator was used to stimulate and activate root system development of *Pennisetum* 'Vertigo'. Goteo is a biostimulator composed of marine algae *Ascophyllum nodosum* (source of auxins, cytokinins, vitamins, macro- and microelement supporting various metabolic process and eliciting root-growth-promoting activity) with the addition of phosphorus (13% $P_2O_5$), and potassium (5% $K_2O$), which support rooting process [8–10]. Biological activity of algal phytohormones system has been of great interest in recent years [11]. Rayorath et al. [12] tested the addition of *Ascophyllum nodosum* extracts to the growth medium in *Arabidopsis thaliana* cultivation and demonstrated that the use of seaweed extracts led to enhanced root growth and development. Goteo, as an algal based product for agriculture purposes, is recommended as an activator of root growth mainly based on vegetable crop testing [13]. Studies on ornamental plants have rarely been carried out. Positive effects of Goëmar Goteo on rooting was reported for *Physocarpus opulifolius* [14], *Hydrangea paniculata* [15], *Ornithogalum arabicum* [16] and *Rosa* [17]; however, there are no reports available on the effect of such a biostimulator on the rooting of grass cuttings. We hypothesize that the biostimulator Goteo improves the 'Vertigo' rooting process and the quality of the shoots. The uptake of water and nutrients is the most important function of plant root system [18] and the nutritional status of plants depends not only on the amount of available nutrients in soil, but also on the ability of roots to explore nutrient supplies [19]. Deeper and vigorous roots determine more efficient acquisition of mineral nutrients from the soil [20], causing enhanced growth of overground parts of plants. Thus, the use of appropriate cultivation techniques during the nursery period results in obtaining the final product with optimal morphological features and size (roots and aerial parts), which in consequence is essential for plant establishment and subsequent growth after transplantation to various environmental field conditions [21].

*Pennisetum* 'Vertigo' is currently gaining an evident attention in large-scale production, but propagation methods used by the producers are still inefficient and scientific recommendations have not been published anywhere. This study aimed to evaluate cutting rooting of *Pennisetum* 'Vertigo' treated with a biostimulator and grown in two different media. Additionally, the subsequent greenhouse development and nutritional status of plants obtained from cuttings were analyzed until the final product was achieved. The obtained results may expand basic knowledge concerning propagation and growth of *Pennisetum* 'Vertigo' plants and they have also an applicative character.

## 2. Materials and Methods

### 2.1. Plant Material

The research material was *Pennisetum* 'Vertigo' cuttings originating from Africa (nursery in Kenia) and imported for research by Plantpol Zaborze (Oświęcim, Poland) on 26 January 2019. The experiment was carried out in a glass-glazed Venlotype greenhouse (equipped with Integro 724 process computers, Priva, Netherlands) at the Faculty of Biotechnology and Horticulture of the University of Agriculture in Kraków, Poland (lat. 50.08° N, long. 19.95° E). Weekly average, lowest and highest day and night temperatures as well as radiation during the experiment, i.e., rooting cuttings (4 weeks) and subsequent cultivation (8 weeks) are presented in Figure 1.

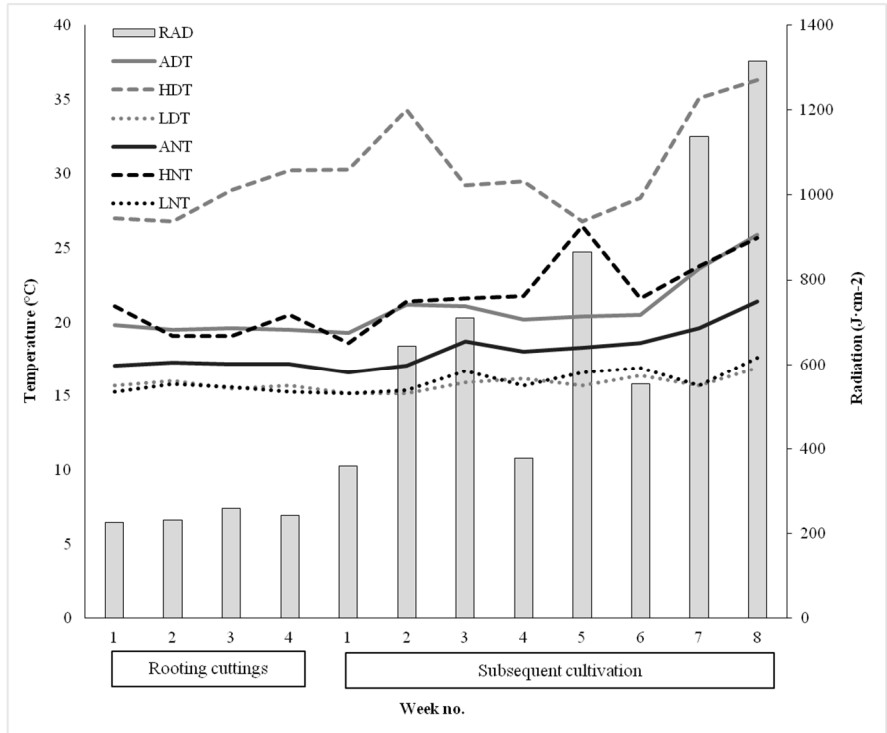

**Figure 1.** Weekly radiation (RAD) as well as weekly average day temperature (ADT), highest day temperature (HDT), lowest day temperature (LDT), average night temperature (ANT), highest night temperature (HNT), lowest night temperature (LNT) during the experiment.

The cuttings were made from the lower part of the stem with the most basal node; the average length of the cuttings was 20 cm (Figure 2A). Before the experiment, single dried leaves and roots from the basal node of the cuttings were removed (no live roots were observed on the cuttings). The cuttings were placed (with the base of the cutting about 2 cm below the medium surface) in openwork plastic boxes with a mesh bottom (32 cm × 45 cm × 8 cm) filled with two different media: perlite (fraction 2–6 mm) or peat (Klasmann TS2, Klasmann-Deilmann GmbH, Germany) (Figure 2B). Forty cuttings were placed in each box. For each medium, the following biostimulator Goteo (Goëmar Lab. Company, France) treatments at a concentration of 0.1% were used:

- soaking the 2 cm cuttings base for 20 min in Goteo before the cutting were stuck (soaking),
- watering the cuttings with Goteo 5 times every 5 days (watering),
- applying foliar Goteo spraying 5 times every 5 days (spraying),
- for each Goteo treatment, a control was set up in which the cuttings were treated as follows: soaking with water, watering with water and spraying with water. An absolute control trial was also set up in which the cuttings were not treated with either the Goteo or the water; the results obtained for the absolute control were not statistically different from those obtained with the 3 control methods treated with water. On this basis, only the absolute control was selected for further statistical analyzes (control).

Boxes with cuttings were placed in a greenhouse under natural light conditions and were covered with perforated plastic film (125 holes/m$^2$) to provide optimal moisture conditions. After 7 days, the film was gradually opened, and it was completely removed after 14 days.

Biometric observations and measurements were made 4 weeks after the beginning of the experiment. Cutting growth was evaluated, taking into consideration the following features: percentage of rooted cuttings, number of roots per cutting, the average length of a single root, fresh weight (FW) of roots and root dry weight (DW) by drying at 105 °C.

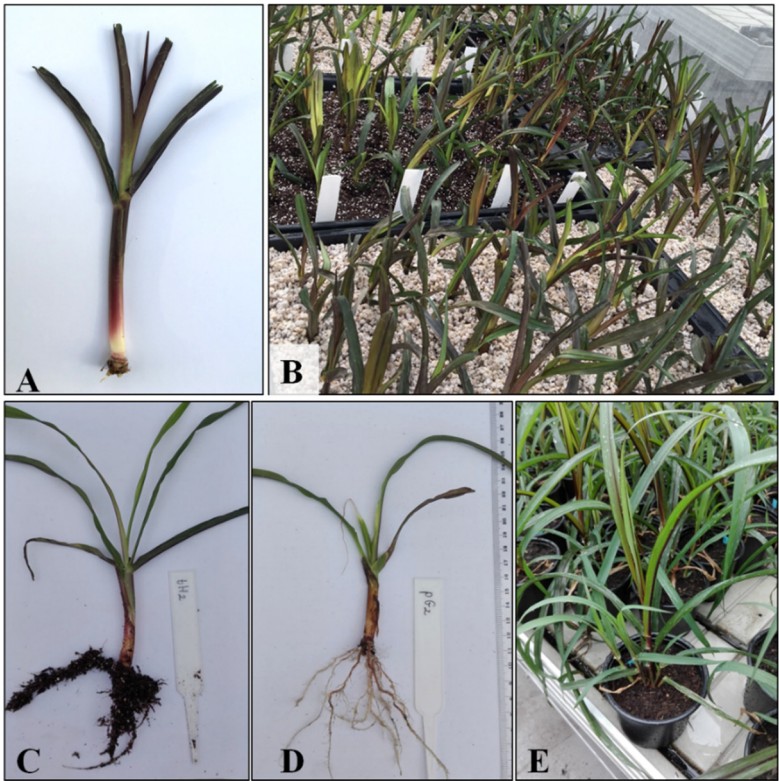

**Figure 2.** (**A**) prepared cutting, (**B**) cuttings placed in two different media: perlite or peat, (**C**) cutting rooted in peat, (**D**) cutting rooted in perlite, (**E**) final product obtained after 2 months of pot cultivation.

## 2.2. Subsequent Cultivation to the Final Product

Then, for further observations of growth and development, the rooted cuttings were planted individually in plastic pots (12 cm in diameter and a volume of 0.6 dm$^3$) filled with peat substrate (Klasmann TS2, Klasmann-Deilmann GmbH, Germany). The plants were cultivated in the same greenhouse compartment as during the rooting period and in an identical experimental arrangement as during the rooting period, i.e., plants rooted in perlite or peat and previously treated with Goteo (plus control) were separate experimental objects. Top-dressing was not applied, and the frequency of irrigation was adjusted to the plant growth phase. After 2 months of pot cultivation, the plant height (from the base of the stems to the uppermost part of the newly formed leaf), number of stems per plant and fresh weight (FW) of aerial parts (aboveground part) were evaluated.

## 2.3. Nutritional Status of Plants

Chemical composition of leaves was investigated. The leaves were sampled randomly; there were 5 leaves without drying symptoms from each plant. A quarter of the leaves were chopped with a knife and dry weight (DW) was assayed at 105 °C. The remaining leaves were dried at 65 °C (24 h) in forced air oven. Dried samples were grinded in a laboratory grinder (FRITSCH Pulverisette 14, Germany) using a 0.5 mm sieve and P, K, Ca, Mg, Cu, Fe, B, Mn, Mo and Zn content was determined after mineralization in 65% extra pure HNO$_3$ in a CEM MARS-5 Xpress (CEM World Headquarters, Matthews, NC, USA) microwave system [22], using an ICP-OES high-dispersion spectrometer (Leeman Labs, New Hampshire, MA, USA). Leaf N content was assayed by the Kjeldahl method using a UDK 193 Distillation Unit (VELP Scientifica, Inc. Bohemia, NY, USA) [23]. Four analytical replications were performed for each treatment.

*2.4. Statistical Analysis*

Rooting experiment was set up with 4 replications of each treatment combination: 2 types of media × 4 Goteo treatments (including control). Each replication consisted of ten cuttings. In total, 320 plants were tested in this experiment. In the second part of the study (subsequent cultivation), there were 20 plants in each treatment (4 replicates, 5 plants each); in total 180 plants were analyzed.

All data were analyzed using STATISTICA version 10.0 data analysis software system (StatSoft, Tulsa, OK, USA). Experimental data were subjected to two-way analysis of variance and then to Tukey's multiple range test to separate the means at the significance level of $p \leq 0.05$. In addition, correlations between analyzed plant features were tested at a probability $p \leq 0.05$ using the Pearson correlation coefficient.

**3. Results**

*3.1. Rooting of Cuttings*

*Pennisetum* 'Vertigo' cuttings rooted at a level of 83.3 to 100.0% (Table 1). Considering only the medium, it was found that perlite was better for rooting the cuttings (95.8%) compared to peat (90.3%), while assessing the effect of Goteo, regardless of the medium, it was found that spraying and watering stimulated the rooting process (97.2 and 100.0%, respectively). The best rooting results (100%) were obtained in perlite when cuttings were sprayed and Goteo-watered, and also in peat when they were Goteo-watered; these results were 6% (perlite) and 16% (peat) better compared to control.

**Table 1.** Effect of medium and biostimulator Goteo on the rooting percentage and root parameters of *Pennisetum* 'Vertigo' cuttings.

| Medium | Goteo | Rooting (%) | Roots (no.) | Root Length (cm) | Root FW (g) | Root DW (%) |
|---|---|---|---|---|---|---|
| Peat | Control | 83.7 ± 5.5 c | 2.3 ± 0.5 c | 5.7 ± 1.1 e | 0.7 ± 0.1 b | 9.2 ± 0.5 a |
| | Soaking | 83.3 ± 5.5 c | 2.5 ± 0.2 bc | 6.7 ± 0.7 de | 0.8 ± 0.1 b | 8.5 ± 0.1 ab |
| | Spraying | 94.3 ± 5.5 ab | 2.4 ± 0.4 bc | 7.2 ± 0.6 de | 0.7 ± 0.1 b | 8.4 ± 0.4 ab |
| | Watering | 100.0 ± 0.0 a | 2.6 ± 0.2 bc | 8.3 ± 0.8 c | 0.8 ± 0.1 b | 8.1 ± 0.3 b |
| Perlite | Control | 94.0 ± 5.5 bab | 3.1 ± 0.2 ab | 8.3 ± 1.0 cd | 1.0 ± 0.1 ab | 8.3 ± 1.1 ab |
| | Soaking | 89.0 ± 0.0 bc | 3.0 ± 0.3 ab | 9.6 ± 1.0 bc | 1.2 ± 0.2 a | 8.6 ± 0.2 ab |
| | Spraying | 100.0 ± 0.0 a | 3.2 ± 0.4 a | 10.1 ± 1.3 b | 1.0 ± 0.4 ab | 9.0 ± 0.1 ab |
| | Watering | 100.0 ± 0.0 a | 3.2 ± 1.1 a | 11.9 ± 0.6 a | 1.0 ± 0.1 ab | 8.2 ± 0.4 b |
| Peat | | 90.3 ± 8.4 b | 2.5 ± 0.3 b | 7.0 ± 1.2 b | 0.7 ± 0.1 b | 8.6 ± 0.5 |
| Perlite | | 95.8 ± 5.3 a | 3.3 ± 0.5 a | 10.0 ± 1.6 a | 1.1 ± 0.2 a | 8.5 ± 0.6 |
| | Control | 89.0 ± 7.6 b | 2.7 ± 0.7 | 7.0 ± 1.7 c | 0.8 ± 0.2 | 8.8 ± 0.9 |
| | Soaking | 86.2 ± 4.7 b | 3.0 ± 0.6 | 8.2 ± 1.8 b | 1.0 ± 0.3 | 8.6 ± 0.2 |
| | Spraying | 97.2 ± 4.7 a | 2.8 ± 0.6 | 8.7 ± 1.8 b | 0.9 ± 0.3 | 8.7 ± 0.4 |
| | Watering | 100.0 ± 0.0 a | 2.9 ± 0.8 | 10.1 ± 2.1 a | 0.9 ± 0.1 | 8.2 ± 0.3 |
| **Main Effects** | | | | | | |
| Medium × Goteo | | * | * | * | * | * |
| Medium | | * | * | * | * | NS |
| Goteo | | * | NS | * | NS | NS |

Mean values ± SD in columns followed by different letter(s) are significantly different at $p \leq 0.05$. * Significant effects ($p \leq 0.05$), NS—not significant.

Rooting media had a similar effect on the number, length and fresh mass of adventitious roots formed at the base of the cuttings (Table 1). An average of 3.3 roots, 10 cm long, weighing 1.1 g were recorded on one cutting rooted in perlite, while the roots formed in peat had about 30% lower parameters (Table 1, Figure 2C). A positive effect of Goteo treatment on the length of formed roots was found. In the control it was 7.0 cm, while when treated with soaking and spraying, the roots were longer by more than 1 cm, but watering was the most beneficial, because the cuttings produced the longest roots (10.1 cm). This was confirmed by the results of two-factor statistical analysis, where the

longest roots (11.9 cm) were obtained in the Goteo watering × perlite combination (Table 1, Figure 2D). There was no significant effect of the studied factors on root DW, which was 8.1–9.2%.

### 3.2. Subsequent Cultivation to the Final Product

No effect of the type of substrate in which the cuttings were rooted on the quality of the aerial plant parts, estimated height of plants and number of stems was observed after two months of pot cultivation in the greenhouse (Figure 2E, Table 2). However, the positive effect of Goteo, regardless of its application method, on plant heights (108.1–114.2 cm), which were 21–27 cm higher compared to control (87.0 cm), has been preserved. A particularly beneficial effect of Goteo watering used during rooting of the cuttings was noted, which at this stage of cultivation stimulated stem formation (plant branching) in cuttings rooted in perlite and in peat (4.8 and 4.0 new shoots to 3.2 and 2.2 in control, respectively). Peat-rooted plants after 2 months of pot cultivation had a higher FW aboveground compared to perlite-rooted plants, while those treated with Goteo, and especially watered, increased the FW of aerial plant parts, which was also documented by the results of two-factor analysis. In our study, positive significant correlations were observed for selected analyzed features and they are presented in Table 3.

**Table 2.** Effect of medium and biostimulator Goteo on growth parameters of *Pennisetum* "Vertigo" plants after 2 months of pot cultivation.

| Medium | Goteo | Plant Height (cm) | Stems (no.) | Overground Part FW (g) |
|---|---|---|---|---|
| Peat | Control | 91.0 ± 2.5 c | 3.2 ± 0.1 bc | 47.9 ± 6.6 ab |
| | Soaking | 109.6 ± 7.8 b | 2.8 ± 0.3 bc | 47.4 ± 2.2 ab |
| | Spraying | 105.3 ± 8.6 b | 3.0 ± 0.5 bc | 42.4 ± 5.7 bc |
| | Watering | 107.7 ± 1.7 b | 4.0 ± 0.7 ab | 52.0 ± 6.8 a |
| Perlite | Control | 82.9 ± 1.6 c | 2.2 ± 0.2 c | 39.9 ± 6.2 bc |
| | Soaking | 106.5 ± 6.5 b | 2.2 ± 0.1 c | 36.8 ± 3.9 c |
| | Spraying | 112.8 ± 6.4 ab | 2.2 ± 0.1 c | 42.7 ± 0.4 bc |
| | Watering | 120.8 ± 2.7 a | 4.8 ± 0.2 a | 50.7 ± 4.6 a |
| Peat | - | 103.4 ± 9.0 | 3.2 ± 0.5 | 47.4 ± 6.0 a |
| Perlite | | 105.8 ± 15.3 | 2.8 ± 0.6 | 42.5 ± 6.5 b |
| | Control | 87.0 ± 4.8 b | 2.8 ± 0.3 b | 44.0 ± 7.3 b |
| | Soaking | 108.1 ± 6.7 a | 2.6 ± 0.3 b | 42.1 ± 6.5 b |
| | Spraying | 109.1 ± 7.9 a | 2.6 ± 0.4 b | 42.6 ± 3.6 b |
| | Watering | 114.2 ± 7.5 a | 4.4 ± 0.5 a | 51.4 ± 5.3 a |
| Main Effects | | | | |
| Medium × Goteo | | * | * | * |
| Medium | | NS | NS | * |
| Goteo | | * | * | * |

Mean values ± SD in columns followed by different letter(s) are significantly different at $p \leq 0.05$ * Significant effects ($p \leq 0.05$), NS—not significant.

**Table 3.** Correlation coefficients between analyzed features, irrespective of the medium and biostimulator Goteo.

| | Root Length (cm) | Root FW (g) | Root DW (%) | Plant Height (cm) | Stems (no.) | Aboveground Part FW (g) |
|---|---|---|---|---|---|---|
| Roots (no.) | 0.556 * | 0.539 * | NS | NS | NS | NS |
| Root length (cm) | - | 0.594 * | NS | 0.545 * | NS | NS |
| Root FW (g) | - | - | NS | NS | NS | NS |
| Root DW (%) | - | - | - | NS | NS | NS |
| Plant height (cm) | - | - | - | - | NS | NS |
| Stems (no.) | - | - | - | - | - | 0.421 * |

* Correlations significant at $p \leq 0.05$, NS—not significant.

### 3.3. Nutritional Status of Plants

There was no effect of rooting medium (substrate) on macro-and micronutrient contents in the aerial parts, with the exception of higher content of Fe and Cu in perlite-rooted cuttings, irrespective of Goteo treatment (Tables 4 and 5).

**Table 4.** Mineral composition of *Pennisetum* 'Vertigo' leaves after 2 months of pot cultivation: macroelements (% DW).

| Medium | Goteo | N | P | K | Ca | Mg | S |
|---|---|---|---|---|---|---|---|
| Peat | Control | 1.38 ± 0.04 a | 0.48 ± 0.03 c | 2.42 ± 0.06 b | 0.33 ± 0.03 | 0.40 ± 0.05 | 0.15 ± 0.03 |
| | Soaking | 0.99 ± 0.06 ac | 0.40 ± 0.02 c | 2.23 ± 0.12 b | 0.20 ± 0.02 | 0.29 ± 0.04 | 0.11 ± 0.01 |
| | Spraying | 1.33 ± 0.18 ab | 0.55 ± 0.01 bc | 2.78 ± 0.03 b | 0.26 ± 0.03 | 0.42 ± 0.09 | 0.16 ± 0.02 |
| | Watering | 1.07 ± 0.05 ac | 0.81 ± 0.11 a | 2.72 ± 0.37 b | 0.24 ± 0.04 | 0.44 ± 0.02 | 0.17 ± 0.05 |
| Perlite | Control | 1.13 ± 0.04ac | 0.45 ± 0.01c | 2.19 ± 0.04b | 0.22 ± 0.03 | 0.26 ± 0.02 | 0.75 ± 0.03 |
| | Soaking | 0.94 ± 0.02 bc | 0.61 ± 0.02 ac | 3.68 ± 0.16 a | 0.25 ± 0.02 | 0.39 ± 0.02 | 0.14 ± 0.00 |
| | Spraying | 0.87 ± 0.09 c | 0.45 ± 0.01 c | 2.40 ± 0.05 b | 0.21 ± 0.01 | 0.39 ± 0.02 | 0.13 ± 0.00 |
| | Watering | 1.39 ± 0.08 a | 0.75 ± 0.02 ab | 2.40 ± 0.12 b | 0.26 ± 0.02 | 0.35 ± 0.01 | 0.14 ± 0.01 |
| Peat | - | 1.19 ± 0.07 | 0.56 ± 0.05 | 2.54 ± 0.11 | 0.26 ± 0.02 | 0.37 ± 0.03 | 0.15 ± 0.01 |
| Perlite | - | 1.08 ± 0.07 | 0.57 ± 0.04 | 2.67 ± 0.18 | 0.24 ± 0.01 | 0.39 ± 0.02 | 0.12 ± 0.01 |
| | Control | 1.25 ± 0.06 a | 0.47 ± 0.01 b | 2.31 ± 0.06 b | 0.27 ± 0.03 | 0.33 ± 0.04 | 0.11 ± 0.03 |
| | Soaking | 0.97 ± 0.03 b | 0.51 ± 0.05 b | 2.96 ± 0.34 a | 0.23 ± 0.02 | 0.34 ± 0.03 | 0.13 ± 0.01 |
| | Spraying | 1.10 ± 0.14 ab | 0.50 ± 0.02 b | 2.59 ± 0.08 ab | 0.24 ± 0.02 | 0.40 ± 0.01 | 0.15 ± 0.01 |
| | Watering | 1.23 ± 0.08 a | 0.78 ± 0.05 a | 2.56 ± 0.19 ab | 0.25 ± 0.02 | 0.39 ± 0.05 | 0.16 ± 0.02 |
| Main effects | | | | | | | |
| Medium × Goteo | | * | * | * | NS | NS | NS |
| Medium | | NS | NS | NS | NS | NS | NS |
| Goteo | | * | * | * | NS | NS | NS |

Mean values ±SD in columns followed by different letter(s) are significantly different at $p \leq 0.05$ * Significant effects ($p \leq 0.05$), NS—not significant.

**Table 5.** Mineral composition of *Pennisetum* 'Vertigo' leaves after 2 months of pot cultivation: microelements (mg kg $^{-1}$ DW).

| Medium | Goteo | Fe | B | Cu | Mo | Zn | Mn |
|---|---|---|---|---|---|---|---|
| Peat | Control | 28.37 ± 2.46 b | 5.08 ± 0.67 | 5.67 ± 0.35 b | 2.37 ± 0.27 | 31.51 ± 9.86 | 39.60 ± 3.80 |
| | Soaking | 23.70 ± 2.30 b | 4.19 ± 0.07 | 7.80 ± 0.34 ab | 1.09 ± 0.05 | 21.86 ± 1.50 | 38.44 ± 2.46 |
| | Spraying | 29.42 ± 2.25 b | 5.52 ± 0.44 | 5.82 ± 0.38 b | 1.82 ± 0.15 | 28.17 ± 5.36 | 33.57 ± 5.03 |
| | Watering | 31.65 ± 5.98 ab | 4.12 ± 1.10 | 9.45 ± 1.53 a | 1.48 ± 0.29 | 40.28 ± 10.16 | 52.41 ± 6.30 |
| Perlite | Control | 37.70 ± 3.77 ab | 4.43 ± 0.40 | 7.49 ± 0.32 ab | 1.42 ± 0.11 | 22.99 ± 2.16 | 39.35 ± 4.28 |
| | Soaking | 46.07 ± 1.94 a | 4.66 ± 0.42 | 9.17 ± 0.21 a | 2.03 ± 0.10 | 30.67 ± 2.66 | 43.18 ± 3.10 |
| | Spraying | 26.28 ± 0.78 b | 4.31 ± 0.28 | 9.29 ± 0.22 a | 1.42 ± 0.08 | 28.26 ± 0.46 | 40.10 ± 1.90 |
| | Watering | 30.70 ± 2.04 b | 4.75 ± 0.18 | 8.19 ± 0.16 ab | 1.42 ± 0.08 | 27.20 ± 0.67 | 33.252.08 |
| Peat | - | 28.20 ± 1.77 b | 4.73 ± 0.34 | 7.18 ± 0.57 b | 1.69 ± 0.17 | 30.46 ± 3.81 | 41.00 ± 2.88 |
| Perlite | - | 35.19 ± 2.48 a | 4.54 ± 0.15 | 8.35 ± 0.24 a | 1.57 ± 0.09 | 27.28 ± 1.12 | 38.97 ± 1.68 |
| | Control | 32.87 ± 2.95 | 4.57 ± 0.38 | 6.58 ± 0.46 b | 1.90 ± 0.25 | 27.25 ± 4.90 | 39.47 ± 2.56 |
| | Soaking | 34.88 ± 5.18 | 4.43 ± 0.22 | 8.48 ± 0.36 a | 1.56 ± 0.21 | 26.26 ± 2.40 | 40.81 ± 2.06 |
| | Spraying | 27.85 ± 1.28 | 4.91 ± 0.57 | 7.55 ± 0.80 a | 1.62 ± 0.16 | 28.21 ± 2.40 | 36.84 ± 2.81 |
| | Watering | 31.17 ± 2.83 | 4.43 ± 0.27 | 8.82 ± 0.75 a | 1.45 ± 0.08 | 33.74 ± 5.41 | 42.83 ± 5.21 |
| Main effects | | | | | | | |
| Medium × Goteo | | * | NS | * | NS | NS | NS |
| Medium | | * | NS | * | NS | NS | NS |
| Goteo | | NS | NS | * | NS | NS | NS |

Mean values ±SD in columns followed by different letter(s) are significantly different at $p \leq 0.05$ * Significant effects ($p \leq 0.05$), NS—not significant.

Goteo treatment had no effect on Ca, Mg, S, Fe, B, Mo, Zn and Mn content in plants, irrespective of rooting medium. The lowest N content was found in plants in which roots were soaked in Goteo

($p$ < 0.05). On the other hand, watering resulted in the highest content of P compared to other Goteo treatments. Soaking increased K content which was significantly higher only when compared to the control. Any Goteo treatment increased Cu content in plants compared to the control.

There was a significant interaction between medium and Goteo treatment on N, P, K, Fe and Cu content in plants. There was no clear effect of Goteo treatment on N content in plants which had been rooted in peat, whereas in perlite-rooted plants, watering resulted in the higher N content compared to soaking or spraying. In both rooting media, watering resulted in the highest P content; however, in perlite-rooted plants the differences in P content were significant only between watered and sprayed, and watered and control plants. In all treatments there were similar K contents in plants, except significantly higher K content in perlite-rooted plants which cuttings had been soaked in Goteo. There was no effect of Goteo treatment on Fe content in peat-rooted plants, but in perlite-rooted plants soaking resulted in the highest Fe content. The differences were significant between soaked and watered, and soaked and sprayed plants. In perlite-rooted plants there was no Goteo treatment influence on Cu content, whereas in peat-rooted plants watering of cuttings resulted in the highest Cu content in plants.

## 4. Discussion

Despite the promising results of cutting propagation obtained for selected ornamental grasses [24–26], commercial propagation via stem cuttings has still not been developed for many species and cultivars. Other traditional vegetative methods (e.g., crown division, rhizomes) are more often used in horticultural practice [27–29], but in the case of 'Vertigo', they are not applied because they do not provide sufficient performance.

Grass rooting potential varies and depends on many factors, including genotype [30,31], length of stem-section cuttings [32], cutting position, number of nodes, planting methods [33] and even root-zone temperature [34]. Stem cutting is a fairly efficient multiplication method of many grass species, which satisfies ornamental plant producers. It may be a beneficial alternative to more expensive propagation methods, e.g., in vitro, the use of which is economically justified for perennials [35] rather than for seasonal plants.

The experiment rooting of 'Vertigo' stems was first investigated with the use of two rooting media and Goteo biostimulator during the rooting of cuttings. Perlite has been proven to be a better rooting medium compared to the tested peat. Perlite is extensively used in horticulture to maintain optimal aeration, drainage and moisture that improve environmental conditions in the root growth zone [36]. Therefore, biometric parameters of 'Vertigo' rooting were 30% higher than roots formed in peat. Before all, in perlite rooting of cuttings not treated with biostimulator increased by 10 percent compared to their counterparts rooted in peat.

The choice of rooting growing medium is especially important for producers that propagate new plant varieties. They face a decision: organic versus inorganic rooting medium; should one choose the one that results in a better rooting effect? Perlite is inert and pathogen-free, but it is not a biodegradable inorganic substrate [37]. The fact that it does not decompose may cause the producers to use peat more often. However, it is worth mentioning that perlite can also be reused in commercial plant production after various sterilization methods [38]. Cunliffe et al. [24], studying the rooting of *Pennisetum* 'Rubrum', found that cuttings in peat and perlite rooted at the same level (85%) and rooting in sand was slightly worse (83.3%), while it was the weakest in vermiculite (63.3%). In the experiment with *Pennisetum* 'Vertigo', 100% rooted cuttings were also obtained using peat, but only when Goteo biostimulator was used. Pacholczak and Nowakowska [17] noticed a positive effect of biostimulators on the percentage of rooted cuttings and growth of new shoots in cuttings of ground cover roses. Salachna et al. [16] reported that seaweed extracts stimulated the number and length of roots formed by *Ornitogalum* adventitious bulbs. Similar results were obtained in the 'Vertigo' experiment. Roots formed in perlite were almost 3 cm longer than in the control (without Goteo

treatment), and in peat they were more than 3 cm longer compared to the control. No effect of the tested combinations on the dry root mass was recorded.

Pacholczak and Nowakowska [17] noticed not only the positive effect of biostimulators on rooting but also an increase of chlorophyll and total soluble sugars levels in rose cuttings. According to Agulló-Antón et al. [39], an increase in plant pigments was closely associated with current photosynthesis, which strongly affects subsequent root differentiation. They also argued that auxin application increased sugar accumulation in leaf tissues of rooting cuttings. All this translates into the quality of the plants that are produced.

To obtain and maintain the quality of new products, it is necessary to focus on strengthening the initial stage of the entire plant production process, i.e., on adopting the most optimal and proven propagation method. According to Atkinson [40], cutting reproduction method provides a uniform final product that is easier to manage at the stage of production and sales, because the plants are at the same phase of growth and development. The final *Pennisetum* 'Vertigo' product grown from Goteo-watered cuttings during rooting had the best biometric parameters, plants grown from cuttings rooted in peat had an overground FW higher than those rooted in perlite and it resulted in higher plant quality and greater ornamental value.

More Fe and Cu were detected in plants grown from cuttings rooted in perlite than in peat. Perlite in combination with Goteo soaking increased Fe content, and in both media, there was a tendency for higher Cu content in plants when Goteo was applied. The reason for higher Cu and P and K contents in plants could be the positive effect of Goteo on the development of the plant root system. Higher mass of the root system, overgrowing the entire volume of the substrate, is important in container cultivation, i.e., with a limited amount of substrate, which in this experiment amounted to 0.6 dm$^3$ per one plant. In our study the plants which were watered, soaked or sprayed by Goteo produced a higher number of longer roots. It can be assumed that these plants took up more P and this stimulated the growth of the root system [41]. It can also be assumed that the mechanism of the beneficial effect of Goteo on the development of the 'Vertigo' root system was caused by the presence of a natural source of P in its composition (13% $P_2O_5$). Khan et al. [9] observed a beneficial effect of plant P nutrition as a result of the application of biostimulants based on algae. According to these authors, the development of the root system is also affected by amino acids, polysaccharides, vitamins, and especially auxins present in biostimulants produced on the basis of algae. The increased K content in 'Vertigo' plants rooted in perlite and subjected to Goteo soaking could also be caused by the presence of this component in biostimulant composition. However, this did not correspond to the demonstrated lack of Goteo effect on K uptake applied by watering or spraying the plants. Chouliaras et al. [42] showed an increase in the content of K and Cu in olive leaves, as a result of foliar application of *Ascophyllum nodosum* extract at a concentration of 0.5%. The authors suggested that the influence of seaweed on Cu uptake could be due to the increasing membrane permeability of cells and hormone-like activities of the seaweed caused by their involvement in cell respiration, photosynthesis and various enzymatic reactions. This mechanism could also occur in our experiment, where the application of Goteo generally increased Cu content in plants.

The lack of a clear effect of Goteo on N content in plants in our study could have been caused by a too low biostimulator concentration, because the stimulating effect of red algae depends on the concentration of seaweed extract [43]. There was only a tendency of a higher N content in plants rooted in perlite and watered with Goteo. Similarly, Dudaš et al. [44] showed the lack of influence of watering with algae extract (Bio-algeen) on N content in lettuce. Chouliaras et al. [42] demonstrated a different stimulating effect of biostimulants from marine algae on N uptake and other nutrients in olive, and Rathore et al. [43] did this in soybean.

## 5. Conclusions

For the first time, research on *Pennisetum* 'Vertigo' was carried out that considered the needs of the producers who expect to obtain a satisfactory reproductive rate and plants of the highest

quality. The production method obtained in our study is recommended. The developed procedure for rooting 'Vertigo' cuttings in perlite or peat using 0.1% Goteo watering was effective in 100%, but root parameters, i.e., number of roots, root length and fresh weight, were higher (by 30%) for perlite-rooted cuttings. Goteo biostimulator treatments stimulated the length of formed roots and the height of plants in further cultivation. It is suggested to use either of the media under study in combination with Goteo watering. In the latter combination, the final 'Vertigo' product increased the aboveground weight of plants and their branching which is associated with the best plant nutritional status.

**Author Contributions:** Conceptualization, A.K., B.P. (Bożena Pawłowska); methodology, A.K., B.P. (Bożena Pawłowska), I.K., investigation B.P. (Barbara Prokopiuk), I.K.; formal analysis, A.K., B.P. (Bożena Pawłowska), I.K.; visualization, A.K., B.P. (Barbara Prokopiuk), I.K.; writing—original draft, A.K., B.P. (Bożena Pawłowska), I.K., B.P. (Barbara Prokopiuk); writing—review and editing, A.K., B.P. (Bożena Pawłowska), I.K.; funding acquisition, A.K., B.P. (Bożena Pawłowska); resources: B.P. (Bożena Pawłowska), I.K.; supervision, A.K., B.P. (Bożena Pawłowska); All authors have read and agreed to the published version of the manuscript.

**Funding:** This research was supported by the Ministry of Science and Higher Education of the Republic of Poland from subvention funds for University of Agriculture in Krakow.

**Acknowledgments:** The author thanks to Plantpol Zaborze for the plant material and Łukasz Wojtas for the support in data recording.

**Conflicts of Interest:** The authors declare no conflict of interest.

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
