# Peer review of "Rooting Media and Biostimulator Goteo Treatment Effect the Adventitious Root Formation of Pennisetum ‘Vertigo’ Cuttings and the Quality of the Final Product"

_agriculture, doi:10.3390/agriculture10110570_

Round 1

Reviewer 1 Report

Manuscript review ,, Rooting media and biostimulator Goteo treatment effect the adventitious root formation of Pennisetum ‘Vertigo’ cuttings and the quality of final product’’.

The manuscript has focused on the use of biostimulant Goteo and media in ornamental grass Pennisetum cv. Vertigo propagation by cuttings.  I do not see the significant contribution of this experiment to biological science, but this paper can certainly serve as a basis for the development of better protocols for Pennisetum commercial production. Further research on understand what are the causes of biostimulant pleiotropic effects are strongly advisable. For example to determine if the increment in growth it is due cell division, cell expansion, or both, as well as exploring the transcriptomes or proteomes.

The authors need to follow the following instructions to improve this manuscript.

Page 1, Line 12-25 (Abstract)

The author should introduce the reader to the topic.

Page 1 Line 28-73 (Introduction)

The novelty of this work is not clear. The authors should state clearly how this work can improve the current scientific knowledge.

Line 49 – Unclear, What stress reproduction from cuttings causes? It needs to be discussed.

Line – 71 – Why the content of minerals studied?

I find no statement of hypothesis, and therefore no discussion in the conclusions as to whether your work confirmed or refuted your starting hypothesis.

Page 2 (M & M)

Line 89 – Is peat Klasmann TS2 (line 89) designed for propagation applications as well as a growing on medium for small propagule types? It seems to be a peat substrate (line 114)? Peat Klasmann TS2 can be high in salts, which can inhibit root development. The medium should be tested for pH, soluble salt, and nutrient content in a reputable lab for greenhouse testing.

Line 90 – Why Goteo at a concentration of 0.1% used? (pilot exp.?)

Line 92 – What was soluble salt of water used?

Page 4 (Results)

Line 142-215 - The data presented in the tables are quite complex to be read. I would suggest to put the data in figures.

There is too much not significant data (Table 4 – Ca, Mg, S; Table 5 – Mo, Zn, Mn).

Line 208 -  n = ?

Page 8 (Discussion)

The discussion is too long and wordy. The comparison of Pennisetum with Olea, Glycine, Rosa etc., seems to me to be random. A great many experiments have been performed on the effect of biostimulants on crops, but only the relevant ones should be quoted and compared with the presented results.

Line 217-235 – This part should be included in the Introduction 

Line 237-238 – Is this claim really valid in the context of TS2 composition?

Line 259-264 – Chlorophyll and total sugars were not tested in the experiment

Line 274 – Mechanism of Cu and Fe uptake is not clearly explanation

Page 9 (Conclusion)

Line 308 - – What treatment is the best?

Author Response

Reviewer 1

The manuscript has focused on the use of biostimulant Goteo and media in ornamental grass Pennisetum cv. Vertigo propagation by cuttings.  I do not see the significant contribution of this experiment to biological science, but this paper can certainly serve as a basis for the development of better protocols for Pennisetum commercial production. Further research on understand what are the causes of biostimulant pleiotropic effects are strongly advisable. For example to determine if the increment in growth it is due cell division, cell expansion, or both, as well as exploring the transcriptomes or proteomes.

 The authors need to follow the following instructions to improve this manuscript.

 Page 1, Line 12-25 (Abstract)

The author should introduce the reader to the topic. - done

 Page 1 Line 28-73 (Introduction)

The novelty of this work is not clear. The authors should state clearly how this work can improve the current scientific knowledge. - done (we added this statement at the end of Introduction section)

Line 49 – Unclear, What stress reproduction from cuttings causes? It needs to be discussed. - we removed this unclear sentence

Line – 71 – Why the content of minerals studied?  - the effect of the biostimulator was investigated by the mineral nutritional status and the nutritional status of plants is reflected in the decorative value of plants

I find no statement of hypothesis, and therefore no discussion in the conclusions as to whether your work confirmed or refuted your starting hypothesis. - we added the hypothesis.

 Page 2 (M & M)

Line 89 – Is peat Klasmann TS2 (line 89) designed for propagation applications as well as a growing on medium for small propagule types? It seems to be a peat substrate (line 114)? Peat Klasmann TS2 can be high in salts, which can inhibit root development. The medium should be tested for pH, soluble salt, and nutrient content in a reputable lab for greenhouse testing. - peat TS2 is a standardized substrate with a composition established by the producer, it is commonly used for rooting cuttings.

Line 90 – Why Goteo at a concentration of 0.1% used? (pilot exp.?) - our experiment was a pilot, and it accounted the recommendations of the producer of Goteo.

Line 92 – What was soluble salt of water used? - tap water was used, moreover an absolute control trial was set up  - we explained it M&M section. 

Line 142-215 - The data presented in the tables are quite complex to be read. I would suggest to put the data in figures. - in our opinion, it is easier to include large-scale results in a table and easier to analyze because they are compactly located and easier to identify interactions. We prefer the tables in this case.

There is too much not significant data (Table 4 – Ca, Mg, S; Table 5 – Mo, Zn, Mn). - although there are no statistical differences, the obtained numerical values showing the level of these elements are also cognitively important.

Line 208 -  n = ? we added this information into the M&M

 Page 8 (Discussion)

The discussion is too long and wordy - some sentences were removed to Introduction section. The comparison of Pennisetum with Olea, Glycine, Rosa etc., seems to me to be random. A great many experiments have been performed on the effect of biostimulants on crops, but only the relevant ones should be quoted and compared with the presented results. - we referred to ornamental plants in the Discussion, analyzing rooting (Ornithogalum, Rosa), but there is no such research on ornamental plants in the context of the nutritional status.

Line 217-235 – This part should be included in the Introduction - done

Line 237-238 – Is this claim really valid in the context of TS2 composition? - the sentence has been clarified

Line 259-264 – Chlorophyll and total sugars were not tested in the experiment - yes, but in the Discussion we pointed out a wider context of the possible influence of a biostimulator on plant quality, that is way we prefer to leave this sentence.

Line 274 – Mechanism of Cu and Fe uptake is not clearly explanation - there are such information, especially about Cu at the end of the Discussion.

 Page 9 (Conclusion)

Line 308 - – What treatment is the best? - we recommend to use either of the media under study in combination with Goteo watering (we have written this recommendation in Conclusions)

Reviewer 2 Report

The following slight ambiguities and minor linguistic errors were fund:

Verse 14-15: the sentence is unclear and should be re-arranged, for example: 100% rooting was obtained for plants in perlite when Goteo spraying or watering was used while for plants rooted in peat only after Goteo watering application.

Verse 36 and 76: Pennisetum – italics

Verse 51: “macro and microelement” should be “macro- and microelement”

Verse 134: “combination”  used here means “treatment” and this word should replace “combination’

Verse 147, 148: ….Goteo watered, and also in peat  when they were Goteo-watered;  It should be standardized as in the whole text.

Verse 192:  ..  soaked  by Gotem.. Things are soaked in  (the solutions)

Verse 194: “Soaking increased K content but it was significantly higher only when  compared to control.”  It suggests ‘soaking”,   rather “which” (the content) should be used here

Verse  196:   “There was a significant interaction rooting medium × Goteo treatment…”  something is missing here:  rather: interaction between medium and Goteo treatment

Verse 226: via – italics

Verse 284: “ produced longer and higher number of roots’. Maybe more clear: produced higher number of longer roots’

Verse 315: the sentence: “ It is suggested to use both media in combination with watering Goteo…” is not precise as it may suggest mixing both media.  Wouldn’t it be more clear  :  ..to use either  (any) of the media under study in combination with Goteo watering ? Or:  Both media were equally useful when used with Goteo watering ?

Author Response

Reviewer 2

The following slight ambiguities and minor linguistic errors were fund:

Verse 14-15: the sentence is unclear and should be re-arranged, for example: 100% rooting was obtained for plants in perlite when Goteo spraying or watering was used while for plants rooted in peat only after Goteo watering application. - done

Verse 36 and 76: Pennisetum – italics  - done

Verse 51: “macro and microelement” should be “macro- and microelement” - done

Verse 134: “combination”  used here means “treatment” and this word should replace “combination’ -done

Verse 147, 148: ….Goteo watered, and also in peat  when they were Goteo-watered;  It should be standardized as in the whole text. - done

Verse 192:  ..  soaked  by Gotem.. Things are soaked in  (the solutions) - done

Verse 194: “Soaking increased K content but it was significantly higher only when  compared to control.”  It suggests ‘soaking”,   rather “which” (the content) should be used here - done

Verse  196:   “There was a significant interaction rooting medium × Goteo treatment…”  something is missing here:  rather: interaction between medium and Goteo treatment - done

Verse 226: via – italics - done

Verse 284: “ produced longer and higher number of roots’. Maybe more clear: produced higher number of longer roots’ - done

Verse 315: the sentence: “ It is suggested to use both media in combination with watering Goteo…” is not precise as it may suggest mixing both media.  Wouldn’t it be more clear  :  ..to use either  (any) of the media under study in combination with Goteo watering ? Or:  Both media were equally useful when used with Goteo watering ? - done

Reviewer 3 Report

The paper falls well in the aims of Journal and it shows a good degree of novelty and interest for researchers and farmers. The presentation is clear in all parts of the paper. However; I suggest that authors read well the guidelines of the Journal: for example uppercase for the all words of paragraph heads.

Specific comments

Material and methods

Figure 1: Please, add in the capture of figure 1 the description of abbreviations reported in the figure legend (for example: ADT = average daily temperature, tec.)

Results

Line 144: the percentage (98.8%) is wrong, because if the authors do the average of the four values of cuttings grown in perlite, it results in 95.75% and not 98.8%. Please verify.

Tables (1, 2, 4 and 5): the letter “a” for significativity should be attributed to the highest value and then the attribution should continue towards the lower values and no vice versa, as the authors made in the all tables of paper. Moreover, if a parameter was not significant, the letters must no be written (see Root DW, FW and length in table 1); please correct it. Finally, if the interaction is significant, the authors should comment on this one and not the main effect of factors, which should not reported in the table.

Line 160: Do the authors refer to interaction “medium x Goteo” for roots length? If yes, the interaction was not significant and it is incorrect to comment on it.

Figure 2: the letter B isn’t well visible, please correct.

Table 2: For overground part FW, the significativity letter is wrong, please correct it.

Discussion

In some parts of the discussion, the authors report again the results, please verify and correct these parts.

Author Response

Reviewer 3

The paper falls well in the aims of Journal and it shows a good degree of novelty and interest for researchers and farmers. The presentation is clear in all parts of the paper. However; I suggest that authors read well the guidelines of the Journal: for example uppercase for the all words of paragraph heads. - done

Specific comments

Material and methods

Figure 1: Please, add in the capture of figure 1 the description of abbreviations reported in the figure legend (for example: ADT = average daily temperature, tec.) - done

Results

Line 144: the percentage (98.8%) is wrong, because if the authors do the average of the four values of cuttings grown in perlite, it results in 95.75% and not 98.8%. Please verify. - done

Tables (1, 2, 4 and 5): the letter “a” for significativity should be attributed to the highest value and then the attribution should continue towards the lower values and no vice versa, as the authors made in the all tables of paper. Moreover, if a parameter was not significant, the letters must no be written (see Root DW, FW and length in table 1); please correct it. - done

 Finally, if the interaction is significant, the authors should comment on this one and not the main effect of factors, which should not reported in the table. - done (the table was corrected)

Line 160: Do the authors refer to interaction “medium x Goteo” for roots length? If yes, the interaction was not significant and it is incorrect to comment on it. - it was significant, we have verified it in the table 1.

Figure 2: the letter B isn’t well visible, please correct. - done

Table 2: For overground part FW, the significativity letter is wrong, please correct it. - done

Discussion

In some parts of the discussion, the authors report again the results, please verify and correct these parts. - we corrected Discussion according to this suggestion.

Reviewer 4 Report

Dear Authors, you should address my comments highlighted across the text and Tables.

Author Response

Reviewer 4

The pdf attachment titled "peer-review-9321114.v2.pdf"

we took into account all the Reviewer's comments and corrected the manuscript according to them.

Round 2

Reviewer 1 Report

Although the resubmitted manuscript was slightly redesigned, the new version does not add much value to the scientific level of the investigation. The authors think that “peat TS2 is …used for rooting cuttings”. Are they any reports that TS2 is intended for rooting? Quality of medium to root plant from cutting is dependent on multivariate parameters such as pH, physical stability, water absorption and retention, nutrients. This information the authors have not put in the manuscript. According to the authors TS2 is so commonly used  for starting plant cuttings. So what's the trouble? In my opinion, the lack of information what are the composition of media is a serious drawback of manuscript. 

Other remarks:

Table 1 - please, correct the letters indicating the significance of differences between the means (Rooting % - Goteo effect) and (Roots no. - interaction)

Table 2 and 3  – please, determine ,,Overground part’’ or ,,Aboveground part’’  (see text)

Line 291, 332 - correct Pennistetum to Pennisetum

Author Response

Please, see attached file.
